# Use of Few-Layer Graphene Synthesized under Conditions of Self-Propagating High-Temperature Synthesis for Supercapacitors Applications

**DOI:** 10.3390/nano13162368

**Published:** 2023-08-18

**Authors:** Alexey A. Vozniakovskii, Evgenia A. Smirnova, Rostislav V. Apraksin, Sergey V. Kidalov, Alexander P. Voznyakovskii

**Affiliations:** 1Laboratory “Physics for Cluster Structures”, Ioffe Institute, 194021 Saint-Petersburg, Russia; alexey_inform@mail.ru; 2Laboratory of new functional materials for chemical current sources, Ioffe Institute, 194021 Saint-Petersburg, Russia; esmirnova@mail.ioffe.ru (E.A.S.); apraksinrv@mail.ioffe.ru (R.V.A.); 3Institute of Synthetic Rubber, 198035 St. Petersburg, Russia; voznap@mail.ru

**Keywords:** few-layer graphene, self-propagating high-temperature synthesis, supercapacitors

## Abstract

Graphene nanostructures (GNSs) are among the most promising materials for producing supercapacitors. However, GNSs are still not used in creating supercapacitors due to the impossibility of obtaining large volumes of high-quality material at an acceptable cost. In our previous works, we have shown the possibility of synthesizing large volumes of few-layer graphene (FLG, the number of layers is not more than five) from cyclic biopolymers under conditions of self-propagating high-temperature synthesis (SHS). Using the SHS process makes it possible to synthesize large volumes of FLG without Stone–Wales defects. This work is devoted to the study of the possibility of using FLG synthesized under the conditions of the SHS process in the creation of supercapacitors. It was found that the synthesized FLG makes it possible to obtain better results than using classical materials, namely activated carbon (AC). It was found that the sample based on FLG had a higher specific capacitance of 65 F × g^−1^ compared to the sample from AC, the specific capacitance of which was 35 F × g^−1^; for a speed of 5 mV × s^−1^, these values were170 and 64 F × g^−1^, respectively. The drop in capacitance over 1000 cycles was 4%, indicating a sufficiently high FLG stability, allowing us to consider FLG as a prospective material for use in supercapacitors.

## 1. Introduction

Solving the problem of rapidly growing global energy consumption, combined with the critical issue of climate change, is one of the important challenges for scientists and engineers worldwide, requiring the development of new sustainable and renewable energy sources [1]. Energy storage devices, a key solution to the above problems, must be portable, economical, easy to maintain, energy-efficient, and environmentally friendly. However, the development of such devices is becoming increasingly tricky [2,3,4].

Supercapacitors are energy storage devices with high power characteristics, demonstrating a high charge/discharge rate, cycle life, shelf life, a wide operating temperature range, and safety. [5]. General Electric obtained the first patent for an electrochemical capacitor in 1957 [6], and the term “supercapacitor” was introduced by the first manufacturers of electrochemical capacitors—Nippon Electric Company [7]. Since being commercialized about 40 years ago, supercapacitors have become good electrical energy storage devices for various applications including renewable energy, transportation, and portable electronic devices [8,9]. The operation of a supercapacitor is based on the principle of maintaining an electric charge on two opposite electrodes separated by a dielectric, and the accumulation of energy in them occurs due to the accumulation of charge by forming an electric double layer of electrolyte ions on the surface of conductive electrodes [10].

Among all supercapacitors, one can highlight devices based on aqueous electrolytes, which are both high-power, environmentally friendly, and attractive for mass production [11,12,13,14]. The search for new, environmentally friendly, renewable, and cheap materials for supercapacitor electrodes is an urgent task that contributes to creating more efficient devices with high specific, capacitive, and power characteristics [15,16]. Currently, activated carbon (AC), carbon nanotubes (CNT), templated carbon, carbon onion, etc., have been proposed as functional materials for the electrodes of such supercapacitors [17,18,19]. AC is the most used among these materials due to its high availability, large surface area, and low cost. However, oxygen, nitrogen, and other atoms in amorphous AC structures lead to conduction limitation, adversely affecting the supercapacitor’s specific power [20,21]. CNTs have also been actively explored as electrode materials for supercapacitors [21,22]. However, most CNTs are known to be bound to each other by van der Waals forces between individual nanotubes, which hinders electrolyte diffusion. Despite the advantages, the main problem with CNT-based supercapacitors is their cost [20,23]. Other carbon materials such as templated carbon, carbon onion, and carbide-derived carbon can be used effectively in supercapacitors, but are challenging to manufacture, so the cost of such materials is high, which limits their use in end devices [18,23].

The concept of adding graphene to supercapacitors was first described in 2008 [24]. The authors showed that graphene could replace carbon because it offers high electrical conductivity and demonstrates the “exciting potential for high performance, electrical energy storage devices based on this new class of carbon material”. At the moment, graphene and graphene nanostructures (GNS) can be distinguished among the most promising materials for creating supercapacitors [12,25,26], batteries [27], and other areas of electrical engineering [28,29]. Graphene has an exceptional theoretical specific surface (more than 2630 m^2^ × g^−1^) [30], increased mechanical strength (Young’s modulus of single-layer graphene ~1 TPa), is thermophysical (thermal conductivity of single-layer graphene ~5000 W/(m × K)), and has a unique set of electrochemical properties [31,32], distinguishing it from activated carbon, carbon nanotubes, and fullerenes [12,22,33].

The capacitance characteristics of graphene-based supercapacitors depend on key material characteristics such as the specific surface area, pore size distribution, interlayer spacing, heteroatom doping, surface functionalization, and conductivity [34,35]. Most of these parameters strongly depend on each other and are determined mainly by the graphene synthesis method. The production of functional materials for new generations of energy storage devices often requires high-tech equipment and processes that can only sometimes be easily scaled up for mass production, significantly contributing to the cost of the final devices.

According to the global requirements for sustainable materials, some green technologies have been used to synthesize graphene nanostructures for supercapacitor applications. These approaches aim to minimize toxic chemicals, reduce energy consumption, and utilize renewable resources, making graphene synthesis more environmentally friendly and sustainable. The methods typically use renewable and sustainable carbon sources such as biomass [36], plants [37,38], or other bio-based materials such as lignin, rice husks, bark, etc. [39,40]. An essential advantage of such materials is their low cost. In addition, using materials such as lignin and rice husks or bark as feedstock for synthesizing graphene nanostructures makes it possible to solve the critical problem of their processing into a sought-after product.

Green-synthesized graphene can be used in supercapacitors by directly using it as an electrode material [41,42,43] or by incorporating it into composites with other materials [44,45] to enhance electrode performance. The high electrical conductivity and large surface area of GNSs enable efficient charge storage and fast charge/discharge rates, leading to improved supercapacitor performance. Green-synthesized graphene offers a sustainable and environmentally friendly approach to synthesizing graphene for supercapacitor applications, contributing to the development of cleaner energy storage technologies.

However, despite all the promise of using GNSs in creating supercapacitors, their practical application has yet to occur. One of the critical reasons for this is the high cost of GNSs, which makes the use of GNSs unprofitable. The high price of GNSs is due to the imperfection of the GNS synthesis methods, which do not allow for the synthesis of large volumes of high-quality material at an acceptable cost. Considering the methods of the synthesis of GNS, two main approaches can be distinguished: “top–down” [46] and “bottom–up” [47]. In synthesizing graphene nanostructures by the “top–down” method, the synthesis occurs by isolating graphene structures from ready-made materials, primarily graphite. These include the separation of graphene with tape, exfoliating graphite using surface active agents (surfactants) and ultrasonic (US) irradiation, and reducing graphite oxidized by the Hummers method. In synthesizing graphene nanostructures by the “bottom–up” method, the synthesis occurs by assembling graphene structures from carbon-containing clusters (silicon carbide, carbonaceous gas, etc.). This approach includes pyrolysis, chemical vapor deposition (CVD), and epitaxial growth. The main advantage of methods based on the top–down approach is their high productivity, low cost, and relative technical simplicity. However, these techniques do not allow for obtaining few-layer, high-quality graphene, with reproducible properties of graphene structures. In contrast, the “bottom–up” approach makes it possible to synthesize high-quality graphene nanostructures and strictly control their final properties. However, this approach is significantly less productive and more expensive [48]. In addition, several techniques such as CVD or epitaxial growth make it possible to synthesize graphene only in thin films, which does not allow for the use of the synthesized material in areas where the material is required in the form of a powder. Therefore, many scientific groups are looking for new graphene synthesis methods.

In our previous works, we reported on the possibility of synthesizing large volumes of few-layer graphene (FLG) from cyclic biopolymers under the conditions of the self-propagating high-temperature synthesis (SHS) [49], free of Stone–Wales defects [50]. The Stone–Wales defect is a connected carbon ring with five and seven atoms, resulting from a 90° rotation of adjacent carbon atoms about their center. This type of defect is typical for graphene nanostructures and carbon nanotubes. The synthesized FLG showed high efficiency as a modifying additive in creating polymer composites by the digital light processing (DLP) method of 3D printing [51] and creating pyrotechnic compositions [52].

This work is devoted to the study of the possibility of using few-graphene synthesized under the conditions of the SHS process in the creation of supercapacitors.

## 2. Materials and Methods

### 2.1. Synthesis of FLG

FLG particles synthesized by the SHS method were taken as the basis for the electrode. The initial biopolymer (starch, analytical grade, NEVAREACTIV Inc., St. Petersburg, Russia CAS: 9005-84-9 was mixed with the oxidizing agent (ammonium nitrate, analytical grade, NEVAREACTIV Inc., St. Petersburg, Russia, CAS: 6484-52-2) in a 6 to 4 ratio using a “drunk barrel” type homogenizer (apmech-spb-1, Russia) for 15 min (60 rpm). Then, the resulting mixture was placed in a reactor and heated to a temperature of 220 °C (initialization of SHS synthesis). After the completion of the synthesis, the resulting graphene (in the form of a powder) was washed with isopropyl alcohol (analytical grade, NEVAREACTIV Inc., St. Petersburg, Russia) and dried in an oven until the weight loss ceased. The yield of graphene was 40 wt. % by weight of the original starch. The procedure for obtaining FLG is described in detail in [49].

### 2.2. Characterization of FLG

To obtain electronic images of the graphene particles, a TESCAN Mira-3M scanning electron microscope (SEM) (TESCAN, Brno, Czech Republic, accelerating voltage—20 kV) and an FEI Tecnai G2 30 S-TWIN transmission electron microscope (FEI, Hillsboro, Oregon, United States, accelerating voltage—50 kV) were used. To obtain electronic images of graphene particles by the TEM method, FLG was placed in toluene (concentration 0.05 mass. %) and sonicated until a stable suspension was obtained. Then, the resulting suspension was applied to a carbon grid and dried until the complete evaporation of toluene (dried in a drying cabinet VTS-K52-250 (ACTAN, Saint-Petersburg, Russia) at 60 °C). The laser diffraction method (using Mastersizer 2000 device (Malvern Panalytical, Malvern, UK)) was used to measure the linear dimensions of the particles. For measurement purposes, a suspension of FLG particles in water (0.05 mass %) was prepared by sonication and an ultrasonic bath “Sapphire” (50 W, 22 kHz).

X-ray phase analysis (Shimadzu XRD-7000 diffractometer, Shimadzu, Kyoto, Japan, (Cu Kα = 0.154051 nm)) was used to measure the number of layers in the FLG. The scanning rate was 1 deg/min.

Raman spectroscopy (Confotec NR500 instrument (532 nm, SOL Instruments, Belarus)) was used to measure the quality of the FLG. FLG particles deposited from a toluene suspension (concentration of 0.05 mass %) on a silicon substrate were used for the measurements.

The Brunauer–Emmett–Teller (BET) method (ASAP 2020 instrument, Micromeritics, Norcross, USA) was used to measure the specific surface area of the FLG sample. Before measurement, the FLG sample was dried for 2 h at 300 °C under vacuum.

### 2.3. Preparation of Electrodes from FLG

The electrodes were prepared according to the following procedure: graphene or carbon powder (22 mg), 2 mg of poly(vinylidene fluoride) (HSV 900 PVDF, Arkema, France) as a binder, and 0.3 mL of N-methyl-2-pyrrolidone (NMP, EMPLURA^®^, Merck KGaA, Germany) as a solvent were carefully mixed with ultrasound until a homogeneous mass (slurry) was obtained. Next, a commercially available MF-2012 glassy carbon electrode (working surface area 0.07 cm^2^; BASi, USA) was evenly coated with the above suspension using a micropipette (0.7 µL) and then dried at 60 °C for 2 h. The load of electroactive material on each received electrode was 51 µg (0.73 mg × cm^−2^) (without considering the mass of PVDF). As a reference sample, electrodes were made from commercial activated carbon (AC, grade AG-3 (GOST 20464-75), specific surface area 740 m^2^ × g^−1^) using a similar procedure.

### 2.4. Electrochemical Measurement Technique

The electrochemical properties of the prepared electrodes were studied on a VSP multichannel potentiostat-galvanostat (BioLogic Science Instruments, Seyssinet-Pariset France) in a three-electrode electrochemical cell containing a glassy carbon plate (3 cm^2^) as an auxiliary electrode and a standard silver chloride reference electrode MF-2056 (Ag/AgCl (3 M KCl); BASi, USA). All potentials in the work are given relative to the Ag/AgCl reference electrode.

The prepared graphene electrodes were tested as electrodes for supercapacitors using cyclic voltammetry (CV), galvanostatic charge–discharge (GCD), and electrochemical impedance spectroscopy (EIS) techniques in 1 M LiClO_4_ solution (Sigma-Aldrich, 99%) in distilled water in the range potentials from −1 to 0.8 V. CV measurements were carried out at various scan rates in the range from 5 to 200 mV × s^−1^.

### 2.5. Specific Capacitance

The specific capacitance (C_sp_, F × g^−1^) was calculated based on the data obtained by the CV and GCD measurements according to Equations (1) and (2), respectively [53].
(1)$$C_{\mathrm{sp}}=\;\frac{Q}{m\times \Delta V}$$
(2)$$C_{\mathrm{sp}}=\;\frac{I\times \Delta t}{m\times \Delta V}$$
where *Q* is the accumulated charge in coulombs (equal to half the integrated area of the corresponding CV curve); *m* (g) is the mass of the active substance; ∆*t* (s) is the discharge time; ∆*V* (V) is the voltage window; *I* (A) is the current during discharge.

GCD was carried out at specific currents from 0.2 to 5 A × g^−1^ in the potential range (a voltage window range) from −1 to 0.8 V. EIS measurements were carried out in the frequency range (100 mHz–100 kHz) with a sinusoidal potential amplitude of 5 mV rms at open circuit potential (OCP). Before the EIS measurements, the electrodes were preliminarily soaked in the electrolyte for 5 min to ensure stability. All parameters were obtained using EC-Lab V11.02 software.

## 3. Results and Discussion

Figure 1 shows the electronic images of the synthesized FLG sample obtained with the help of a scanning electron microscope (SEM) and transmission electron microscope (TEM).

As can be seen in Figure 1, the synthesized particles had a few-layer structure (Figure 1c) and had linear dimensions up to several tens of microns (Figure 1a,b). Studies were carried out by laser diffraction to clarify the linear dimensions of the graphene particles (Figure 2).

As can be seen from Figure 2, although the sample contained particles with lateral sizes of up to several hundred microns (Figure 2a), the proportion of such particles was small (Figure 2b). As can be seen in Figure 2b, most of the particles had linear dimensions of about one μm.

An X-ray diffraction study was carried out to determine the number of layers in the synthesized FLG sample. The results of the experiment are shown in Figure 3.

Based on the position of the 002 peak and its full width at half maximum (FWHM), using the Scherer formula [54], the crystallite size (L) was calculated as equal to 17.7 Å. The formula N = L/d was used to calculate the number of graphene layers in the sample. In the formula presented, “N” is the number of graphene layers in the sample, d = 3.81 Å is the interplanar distance. As a result of the calculation, it was found that the number of graphene layers in the sample did not exceed 5.

Figure 4 shows the Raman spectrum of the synthesized FLG sample.

As can be seen from Figure 4, the D, G, 2D, and D + G bands were visible in the Raman spectrum of FLG. Such peaks in the Raman spectrum are typical of graphene nanostructures [55]. The intensity ratio of the D and G peaks was 0.89. In [56], the authors associated the mutual superposition of the 2D and D + G peaks in the region of 2500–3500 cm^−1^ with the wavy structure of the samples, the presence of a large number of edges, and the multidirectional superposition of the graphene layers on each other. It is essential to note that the FLG samples synthesized under the conditions of the SHS process did not contain Stone–Wales defects. This was shown by us using a chemical technique described in detail in our previous work [50].

Figure 5 shows the BET measurement curve of the FLG sample.

As can be seen from Figure 5, the obtained sorption isotherm belonged to type 2 according to the IUPAC classification. This isotherm is characteristic of reversible adsorption on nonporous or macroporous adsorbents by the mechanism of polymolecular adsorption. According to the BET theory of the synthesized FLG sample, the specific surface was 220 m^2^ × g^−1^.

Samples made from FLG and activated carbon (AC) were studied by cyclic voltammetry (CV) as model electrodes of supercapacitors in a three-electrode cell in 1 M aqueous LiClO_4_ solution. Figure 6a,b shows the comparative results of the CV of the fabricated samples at scan rates of 200 (Figure 6a) and 5 mV × s^−1^ (Figure 6b) in the range of potential change from −1 to 0.8 V.

For both scan rates, the samples showed an electrical double-layer capacitance behavior (no pronounced peaks).

The specific capacity of the obtained samples was calculated using Equation (1). Figure 5 shows that at a scan rate of 200 mV × s^−1^ and 5 mV × s^−1^, these values were 170 and 64 F × g^−1^, respectively. It should be noted that despite the fact that the specific surface area of the FLG was much smaller than that of AC (220 and 740 m^2^ × g^−1^, respectively), it demonstrated greater efficiency than AC.

Figure 7a,b show the curves of the current density versus electrode potential at potential scan rates of 5–200 mV × s^−1^.

It can be seen from the presented graphs that for both samples, the values of the current densities became larger with an increasing scan rate. At lower potential scan rates, all samples showed curve shapes close to rectangular (rectangular-like shape), which is typical for double-layer behavior. With an increase in the sweep rate, the shape of the CV curves was distorted (fusiform), which is associated with the resistance of electron transfer. Still, the general shape of the curves remained close to rectangular, which indicates the relatively high-power characteristics of the materials.

Figure 8 shows the dependence lg(I_p_)–lg(v) for the FLG and AC. The values of *I*_p_ were used at the plateau around 0.0–0.2 V, where the contribution of ohmic polarization did not have a significant effect on the CV shape.

The slope of this dependence enables the estimation of the limiting stage of the process [57]. The slope (coefficient b) may take values from 0.5, corresponding to the diffusion-controlled process, to 1, corresponding to adsorption. For the AC sample, the coefficient b values corresponded to mixed kinetics (*b* = 0.86), which suggests insignificant limitations of mass transfer in the electrode material. A noticeable contribution of limited mass transfer was observed for the FLG sample only at high values of scan rates (*b* values close to 0.5).

The galvanostatic charge/discharge method was used to study the capacitive properties of the obtained graphene samples. Figure 9 shows the GCD behavior of all electrodes at current densities of 0.2–5 A × g^−1^. All samples were characterized by deviations from the linearity of the charge–discharge curves, especially when discharging low potentials, which is probably associated with side irreversible reduction processes. Most likely, there is slow water splitting with the hydrogen evolution. Although the overpotential of the hydrogen evolution reaction at carbon electrodes reached high values, at potentials around −1.0 V vs. Ag/AgCl and at low scan rates/discharge currents, the process may have an impact [58,59].

The gravimetric capacitances of the electrodes were calculated using Equation (2). The capacitances of the electrodes based on graphene made from FLG and AC at a current of 0.2 A × g^−1^ were 265 and 75 F × g^−1^, respectively. All calculated capacitance values from CV and GCD are presented in Table 1 and Table 2.

The higher FLG efficiency may be due to a more optimal pore distribution since it is well-known that starting from some value of the surface area, its further increase in surface area does not lead to an increase in capacitance due to limitations of diffusion and electrode penetration into the porous system of the material [60]. Additionally, one of the potential reasons for the better performance of FLG can be considered to be the low concentration of Stone–Wales defects, which we have shown earlier [50,51].

To better understand the electrochemical properties of the electrodes, the samples were studied by electrochemical impedance spectroscopy (EIS). Spectra in Nyquist coordinates are shown in Figure 10. The spectra showed a classical shape for porous systems without a pronounced semicircle and a linear section in the low-frequency region [61]. This indicates a sufficiently high conductivity of the samples and the predominant dominance of diffusion kinetics. The main differences between the samples were observed in the slope of the linear dependence in the low-frequency region, which may be due to more significant diffusion limitations for the sample with FLG. Differences between the samples may be due to specific surface area and porosity differences. These findings are in good agreement with the CV measurements at various sweep rates.

The stability of electrode materials based on few-layer graphene was tested by the GCD method in 1 M LiClO_4_ aqueous solution at a current of 2 A × g^−1^ for 1000 cycles. The resulting dependence is shown in Figure 11.

The drop in capacitance for 1000 cycles was 4% (the initial capacitance was 144 F × g^−1^, and after a thousand cycles, it was 138 F × g^−1^), which indicates a sufficiently high stability of the materials, and allows them to be considered for use in supercapacitors. The decrease in capacity may be associated with irreversible reduction processes (slow water splitting and hydrogen evolution as discussed above), which was confirmed by an increase in the values of the Coulomb efficiency with cycling. It is recommended to use a narrower range of potentials to improve stability.

As shown in [62,63,64,65], the decrease in the capacity of supercapacitors based on carbon nanomaterials has a linear dependence on the number of cycles; therefore, at 5000 cycles, one cannot expect a significant decrease in the capacitance of a supercapacitor based on FLG from the initial value.

The capacitance values of our FLG electrode appeared to be better than those of similar electrode materials based on reduced graphene oxide, N-doped graphene, and graphene/carbon nanotube composites (Table 3).

A comparison of the results obtained in this work with the results obtained by other carbon nanomaterials (see Table 3) is evidence of the effectiveness of the approach proposed in this work. It should be noted that the obtained capacitance values were not record-breaking, but the simple scaling of the proposed synthesis method makes it attractive for its practical realization.

## 4. Conclusions

It was experimentally shown that few-layer graphene synthesized under the conditions of self-propagating high-temperature synthesis is a promising material to use in supercapacitors. It was found that despite the lower specific surface area than the activated carbon, few-layer graphene samples had better capacitive parameters. Samples with few-layer graphene also showed a high stability: the drop in capacity over 1000 cycles was less than 4%.

As a result, few-layer graphene has the optimal structure and particle size distribution and thus achieves higher capacitance values. A high capacitance of 265 F × g^−1^ (0.2 mA × g^−1^) was achieved by the few-layer graphene electrode in an aqueous LiClO_4_ electrolyte. The achieved capacitance values were 2–4 times higher than the activated carbon-based electrode with a higher surface area. Furthermore, the few-layer graphene electrode maintained its capacitive performance cycling (1000 cycles). Thus, the findings suggest that the few-layer graphene composite could be a promising electrode material for supercapacitors.

## Figures and Tables

**Figure 1 nanomaterials-13-02368-f001:**
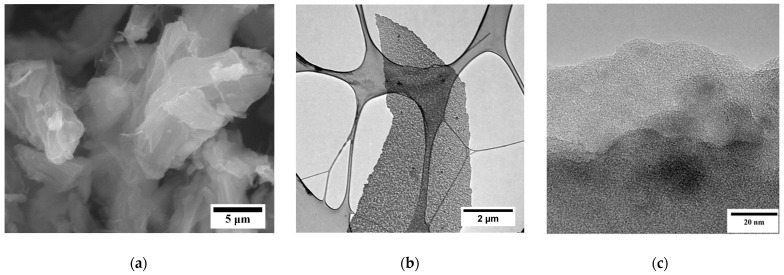
SEM and TEM images of the FLG particles: (**a**) SEM image of FLG; (**b**,**c**) TEM image of FLG.

**Figure 2 nanomaterials-13-02368-f002:**
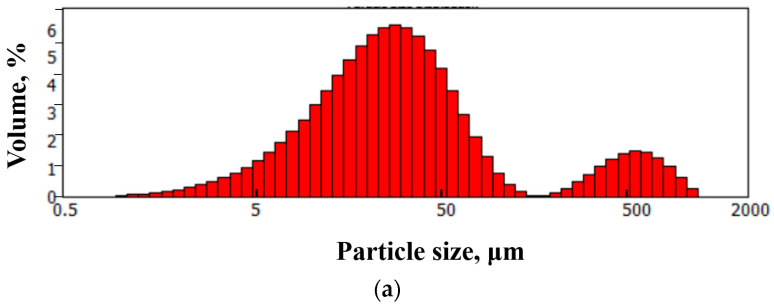
Linear dimensions of the FLG particles. (**a**) Distribution of particle size by volume. (**b**) Distribution of particle size by the number of particles.

**Figure 3 nanomaterials-13-02368-f003:**
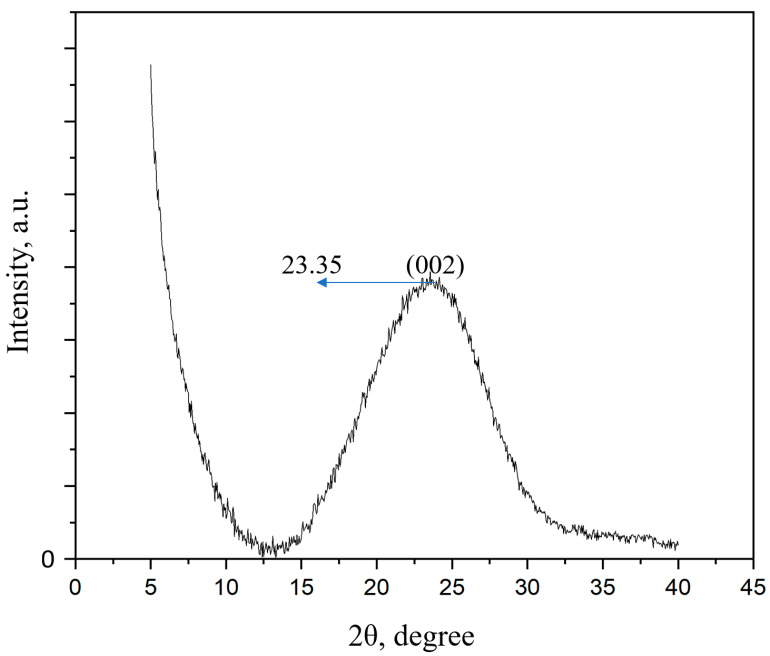
Diffractogram of the FLG sample.

**Figure 4 nanomaterials-13-02368-f004:**
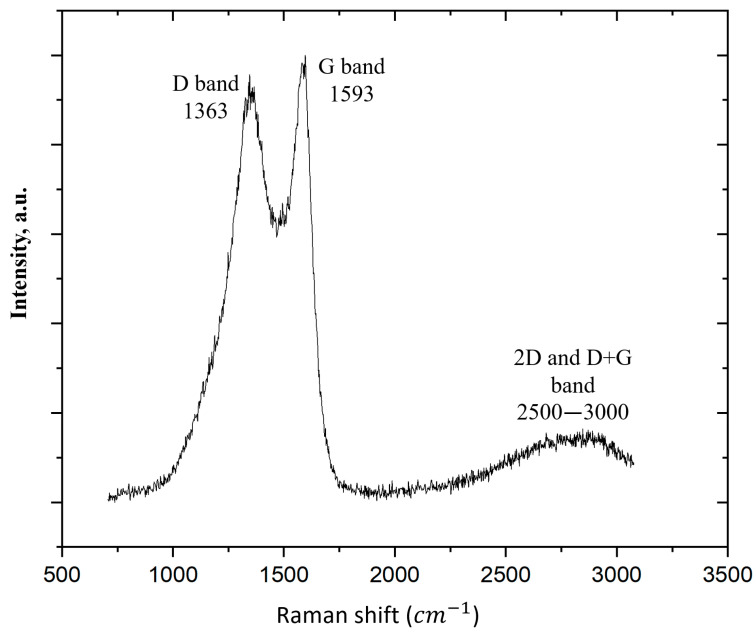
Raman spectra of the FLG sample.

**Figure 5 nanomaterials-13-02368-f005:**
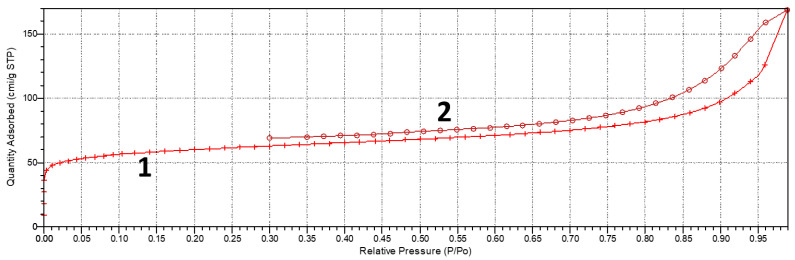
BET measurement curve of the FLG sample. 1—adsorption curve; 2—desorption curve.

**Figure 6 nanomaterials-13-02368-f006:**
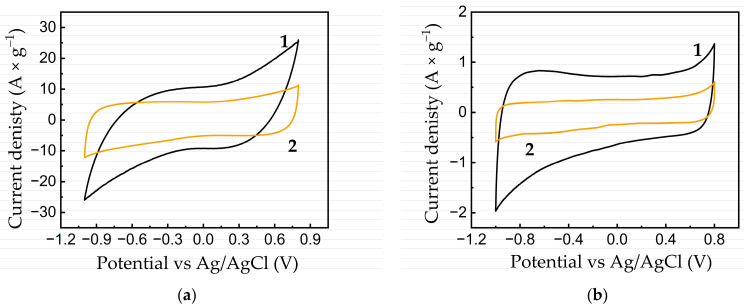
The CV of the FLG (1) and AC (2) samples fabricated in 1 M LiClO_4_ aqueous solution: (**a**) at a potential sweep rate of 200 mV × s^−1^; (**b**) 5 mV × s^−1^.

**Figure 7 nanomaterials-13-02368-f007:**
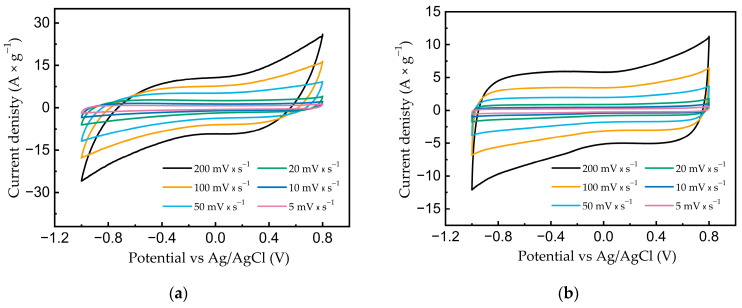
Cyclic voltammograms in 1 M aqueous solution of LiClO_4_ at potential scan rates of 5–200 mV × s^−1^ for (**a**) FLG and (**b**) AC.

**Figure 8 nanomaterials-13-02368-f008:**
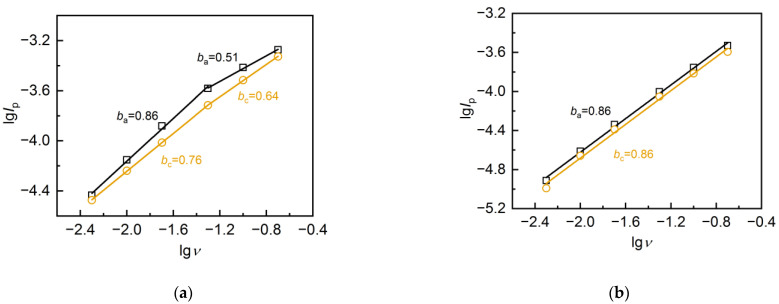
The lg(I)–lg(v) dependencies for (**a**) FLG and (**b**) AC.

**Figure 9 nanomaterials-13-02368-f009:**
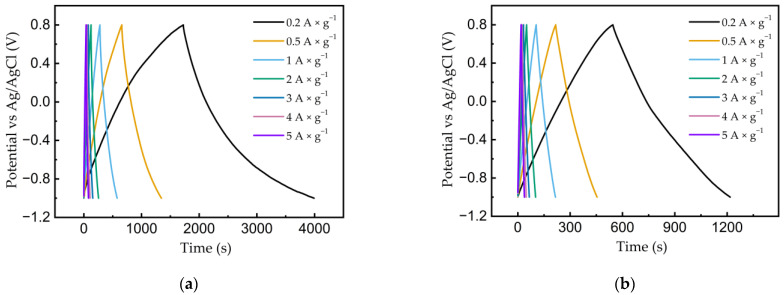
Galvanostatic charge/discharge curves for a graphene sample made in a 1 M LiClO_4_ aqueous solution at current densities of 0.2–5 A × g^−1^ from (**a**) FLG and (**b**) AC.

**Figure 10 nanomaterials-13-02368-f010:**
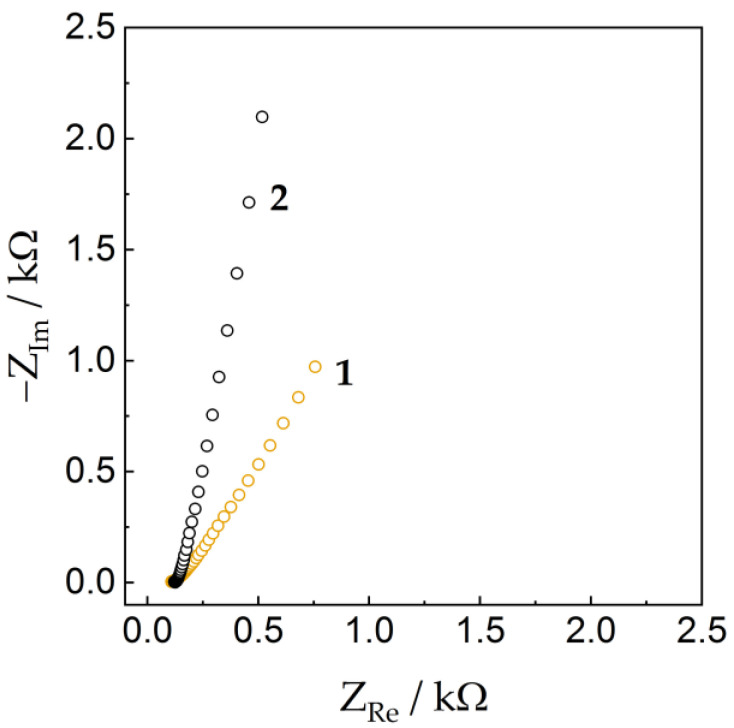
Impedance spectra at OCP (open circuit potential) for electrodes FLG (1) and AC (2).

**Figure 11 nanomaterials-13-02368-f011:**
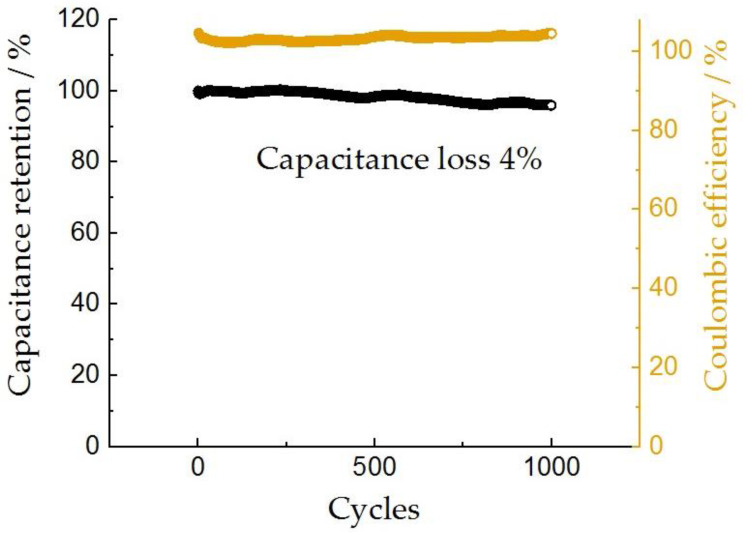
Cycling stability of the FLG electrode measured at a 2 A × g^−1^.

**Table 1 nanomaterials-13-02368-t001:** Capacitance from the CV.

Voltage Scan Rate, mV × s^−1^	Specific Capacitance, F × g^−1^	
	FLG	AC
200	65	35
100	83	38
50	105	43
20	135	50
10	151	57
5	170	65

**Table 2 nanomaterials-13-02368-t002:** Capacitance from the GCD.

Current Density, A × g^−1^	Specific Capacitance, F × g^−1^	
	FLG	AC
0.2	265	75
0.5	194	67
1	168	62
2	145	58
3	130	56
4	121	54
5	113	52

**Table 3 nanomaterials-13-02368-t003:** Comparison of the specific capacitances of the FLG (three-electrode) with those from previous work.

Electrode Materials	Electrolyte	Specific Capacitance, F × g^−1^	References
Few-layer graphene	1 M LiClO_4_	168 (1 A × g^−1^)	This work
N-doped graphene	0.5 M Na_2_SO_4_	≈120 (1 A × g^−1^)	[62]
Carbon/few-layer graphene heterostructure	2 M KOH	≈150 (1 A × g^−1^)	[63]
Plasma-treated graphene oxide	1 M H_2_SO_4_	150 (1 A × g^−1^)	[66]
Three-dimensional graphene/carbon nanotubes	1 M Na_2_SO_4_	≈135 (1 A × g^−1^)	[67]
Reduced graphene oxide	1 M H_2_SO_4_	205 (1 A × g^−1^)	[68]
N-doped graphene	1 M H_2_SO_4_	136 (1 A × g^−1^)	[69]
Reduced graphene oxide	0.5 M H_2_SO_4_	≈105 (1 A × g^−1^)	[70]

## Data Availability

Not applicable.

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
