# Peer review of "Use of Few-Layer Graphene Synthesized under Conditions of Self-Propagating High-Temperature Synthesis for Supercapacitors Applications"

_nanomaterials, 2023, doi:10.3390/nano13162368_

Round 1

Reviewer 1 Report

In this manuscript, Vozniakovskii A.A. et al. investigated the efficiency of using few-layer graphene (FLG) with the number of layers synthesized under self-propagating high-temperature synthesis (SHS) conditions as basic material to be used for supercapacitor applications. This work is interesting for supercapacitor research. Overall this manuscript can be accepted for the publication, but a major revision is absolutely needed. The main points are given below.

  1. Graphene nanostructures (GNS) are among the most promising materials for creating supercapacitors.’ The word ‘creating’ doesn’t seem appropriate here. This word can be replaced by suitable word.
  2. In this work, we investigated the efficiency of using few-layer graphene (FLG) with the number of layers is not more than 5 synthesized under Self-propagating high-temperature synthesis (SHS) conditions as basic material for supercapacitors.’ Authors are advised to simplify this sentence for readers.
  3. Page 1, Line 22-23: ‘for a speed of 5 mV×s-1 these values are 170 and 64 F×g-1, respectively.’  What is speed? It should be scan rate.
  4. Page 1, Line 23: ‘The drop in capacitance over 1000 cycles was 4%, which indicates a sufficiently high stability of FLG and allows us to consider FLG as perspective material for use in supercapacitors.’ only 1000 cycles measurement is not acceptable in current supercapacitor research. The studies covering minimum 5000 cycles and over make semse  for graphene-based electrodes of supercapacitors.
  5. The sub sections from 2.2.1 to 2.2.5 can be written in one section in 2.2 as material characterizations of FLG or similar.
  6. What is ‘∆?’ ’ in equations 1and 2 not mentioned in the text?
  7. Page 7, Line 219-220: ‘The specific surface, according to the BET theory of the synthesized FLG sample, was 220 m2/g.’ BET measurement curve and the nature of the curve should be provided.
  8. Page 7, Line 233-235: ‘It should be noted that despite the fact that the specific surface area of FLG is much smaller than that of AC (220 and 740 m2/g, respectively), it demonstrates greater efficiency than AC.’ Authors can provide more explanation about the better performance of FLG.
  9. Page 7, Line 250: ‘Figure 7 shows the dependence lg(Ip)–lg(v) for FLG and AC.’ Which peak position is considered for this calculation is not mentioned in the text.
  10. Page 9, Line 264-266: ‘All samples are characterized by deviations from the linearity of the charge-discharge curves, especially when discharging to low potentials, which is probably associated with side irreversible recovery processes.’ What do you mean by side irreversible recovery processes? Authors can provide more clear information with references.
  11. In Figure 8. The Y-axis potential should be written as Potential vs Ag/AgCl.
  12. Figure captions of Figure 9 and 10 are same. These must be revised.
  13. Supercapacitor device performance of this electrode material should be provided in order to emphasize the benefits brought by the developed material.
  14. Some related literature works based on SC can be suggested to include. This is essentially needed to support the novelty of this work which lacks at present state. Two examples of these literature are: Chem. Soc. Rev. 2015, 44, 74847539; J. Compos. Sci. 20237(3), 108; International Journal of Energy Research 45 (2021), 16908-16921, etc.

  1. The language and format of the text should be checked carefully.
  2. Full-stop at the end of the title should be removed. Full address details of authors can be provided in the manuscript.

Author Response

Reviewer 1

We are grateful to the reviewer for his attention to our publication. We have corrected all the comments and hope that our publication will be worthy of publication in “nanomaterials”. Corrections are marked in yellow in the text.

Below are the responses to the comments of the reviewer.

  1. Graphene nanostructures (GNS) are among the most promising materials for creating supercapacitors.’ The word ‘creating’ doesn’t seem appropriate here. This word can be replaced by suitable word.

Answer: We have made the necessary changes to the text.

  1. ‘In this work, we investigated the efficiency of using few-layer graphene (FLG) with the number of layers is not more than 5 synthesized under Self-propagating high-temperature synthesis (SHS) conditions as basic material for supercapacitors.’ Authors are advised to simplify this sentence for readers.

Answer: We have made the necessary changes to the text.

  1. Page 1, Line 22-23: ‘for a speed of 5 mV×s-1 these values are 170 and 64 F×g-1, respectively.’  What is speed? It should be scan rate.

Answer: Yes, it should be scan rate. It was a typo. We have updated the manuscript.

  1. Page 1, Line 23: ‘The drop in capacitance over 1000 cycles was 4%, which indicates a sufficiently high stability of FLG and allows us to consider FLG as perspective material for use in supercapacitors.’ only 1000 cycles measurement is not acceptable in current supercapacitor research. The studies covering minimum 5000 cycles and over make semse  for graphene-based electrodes of supercapacitors.

Answer:For this type of materials we can expect a smooth linear decrease in capacitance, without noticeable changes. This is confirmed in many published papers for similar electrode materials (10.1016/j.apsusc.2022.153156 ; 10.1016/j.jelechem.2021.115626 ; 10.1002/admt.202101105 ; 10.1016/j.jallcom.2022.168568). We have made the necessary changes to the text.

  1. The sub sections from 2.2.1 to 2.2.5 can be written in one section in 2.2 as material characterizations of FLG or similar.

Answer: We have revised section 2.2 in accordance with the reviewer's note.

  1. What is ‘∆?’ ’ in equations 1and 2 not mentioned in the text?

Answer: The symbol «'» was mistakenly added. We have updated equations. ∆? is the voltage window (it was 1.8 V both for CV and GCD). This point is discussed later in this section after eq. 1 and 2.

  1. Page 7, Line 219-220: ‘The specific surface, according to the BET theory of the synthesized FLG sample, was 220 m2/g.’ BET measurement curve and the nature of the curve should be provided.

Answer: We have added the necessary data and their discussion.

  1. Page 7, Line 233-235: ‘It should be noted that despite the fact that the specific surface area of FLG is much smaller than that of AC (220 and 740 m2/g, respectively), it demonstrates greater efficiency than AC.’ Authors can provide more explanation about the better performance of FLG.

Answer: Due to restrictions on diffusion and electrolyte access into the pore networks of the material, it is most likely that a larger specific surface area has low effect on performance over a certain value [10.1016/j.cej.2022.137191]. Also one of the potential reasons for the better performance of FLG can be considered to be the low concentration of Stone-Wales Defects , which we have shown earlier (10.3390/nano12040657).

. We have made the necessary changes to the text.

  1. Page 7, Line 250: ‘Figure 7 shows the dependence lg(Ip)–lg(v) for FLG and AC.’ Which peak position is considered for this calculation is not mentioned in the text.

Answer: Since there are no pronounced peaks on CV, we used for calculations the values of currents on the plateau around 0.0 - 0.2 V, where the contribution of ohmic polarization does not have a significant effect on the CV shape. We have made these revisions to the text.

  1. Page 9, Line 264-266: ‘All samples are characterized by deviations from the linearity of the charge-discharge curves, especially when discharging to low potentials, which is probably associated with side irreversible recovery processes.’ What do you mean by side irreversible recovery processes? Authors can provide more clear information with references.

Answer: Most likely there is water splitting with the hydrogen evolution. Although the overpotential of hydrogen evolution at carbon electrodes reaches high values, but at potentials around -1.0 V vs Ag/AgCl and at low sweep rates/discharge currents the process can have an effect [10.1021/acsami.6b14732 ; 10.1021/acsami.7b06727].

  1. In Figure 8. The Y-axis potential should be written as Potential vs Ag/AgCl.

Answer: Yes, we corrected this in the revised version.

  1. Figure captions of Figure 9 and 10 are same. These must be revised.

Answer: We apologize for this oversight. This has been corrected in the new version of the manuscript.

  1. Supercapacitor device performance of this electrode material should be provided in order to emphasize the benefits brought by the developed material.

Answer: We fully agree that measurements in two-electrode cells are important for testing supercapacitor materials, but we were given limited time to prepare the revised version of the manuscript, during which we could not possibly make the required measurements. At the same time, the capacitance of the full device can be expected to be about a factor of 4 smaller, since the material is predominantly double-layer active (10.1002/aenm.201401401)

  1. Some related literature works based on SC can be suggested to include. This is essentially needed to support the novelty of this work which lacks at present state. Two examples of these literature are: Chem. Soc. Rev. 2015, 44, 7484–7539; J. Compos. Sci. 2023, 7(3), 108; International Journal of Energy Research 45 (2021), 16908-16921, etc.

Answer: We have added a relevant references.

  1. The language and format of the text should be checked carefully.

Answer: We have improved the quality of the language and format in accordance with the reviewer's comments.

  1. Full-stop at the end of the title should be removed. Full address details of authors can be provided in the manuscript.

Answer: We have made the necessary changes to the text.

Reviewer 2 Report

In this thesis, the authors studied the efficiency of using a few layers of Graphene (FLG) with no more than 5 layers synthesized under the condition of self propagating high-temperature synthesis (SHS) as the basic material of supercapacitors. It was found that the synthesized FLG makes it possible to obtain better results than using classical materials, namely activated carbon (AC). It was found that the sample based on FLG has a higher specific capacitance – 65 F×g-1 compared to the sample from AC, the specific capacitance of which is 35 F×g-1, for a speed of 5 mV×s-1 these values are 170 and 64 F×g-1, respectively. The drop in capacitance over 1000 cycles was 4%, which indicates a sufficiently high stability of FLG and allows us to consider FLG as perspective material for use in supercapacitors. I believe that publication of the manuscript may be considered only after the following issues have been resolved.

1.     In order to better highlight the advantages of this work, the author needs to provide a table to compare related work.

2.     What is the physical mechanism behind the superior performance of this composite system? It is suggested that the author provide relevant energy band and Electron transfer diagrams for analysis.

3.     The introduction can be improved. The articles related to some applications of graphene and graphene oxide materials should be added such as Micromachines 2023, 14(5), 953; Electronics 2023, 12(12), 2655. The articles related to some applications of energy storage devices should be added such as Energy Technology, 2019, 7(6):57; Chemical Engineering Journal, 2023, 453:139831.

4.     The English expression of the whole article needs to be further improved.

 Minor editing of English language required

Author Response

Reviewer 2

We are grateful to the reviewer for his attention to our publication. We have corrected all the comments and hope that our publication will be worthy of publication in “nanomaterials”. Corrections are marked in yellow in the text.

Below are the responses to the comments of the reviewer.

  1. In order to better highlight the advantages of this work, the author needs to provide a table to compare related work.

Answer: We have added a table to compare related work with the results obtained.

  1. What is the physical mechanism behind the superior performance of this composite system? It is suggested that the author provide relevant energy band and Electron transfer diagrams for analysis.

Answer: Due to restrictions on diffusion and electrolyte access into the pore networks of the material, it is most likely that a larger specific surface area has low effect on performance over a certain value [10.1016/j.cej.2022.137191]. Also one of the potential reasons for the better performance of FNG can be considered to be the low concentration of Stone-Wales Defects, which we have shown earlier (10.3390/nano12040657). We agree that calculation the change of interfacial electron configuration or density of states in FNG would be useful for a more detailed understanding of the changes in electrochemical properties, but we were severely time constrained in preparing an updated version of the manuscript. We will try to provide this type of calculations in the next paper, thanks for your recommendations.

  1. The introduction can be improved. The articles related to some applications of graphene and graphene oxide materials should be added such as Micromachines 2023, 14(5), 953; Electronics 2023, 12(12), 2655. The articles related to some applications of energy storage devices should be added such as Energy Technology, 2019, 7(6):57; Chemical Engineering Journal, 2023, 453:139831.

Answer: We have expanded the introduction and used references recommended by the reviewer.

  1. The English expression of the whole article needs to be further improved.

Answer: We have improved the quality of the language in accordance with the reviewer's comments.

Reviewer 3 Report

The author investigated the efficiency of using few-layer graphene with the number of layers is not more than 5 synthesized under self-propagating high-temperature synthesis conditions as basic material for supercapacitors. The topic of this manuscript is of interest to the readers and the experiments are well designed. However, major revision is required before acceptation.

1.       The cited literature in the introduction section is mostly old, please add some latest literature to support it. Some typical references are suggested, e.g. Nanoscale 2022, 14, 8216; Materials & Design 2023, 229, 111904; Journal of Bioresources and Bioproducts 2022, 7 (4), 245-269.

2.       Please introduce the sources of raw materials and reagents used in the experiment.

3.       It is recommended to add more SEM or TEM images to the article. The SEM image does not show a very intuitive few-layer structure.

4.       Please unify the position for labeling in Figure 1 a and b.

5.       Label the vertical and horizontal coordinate scales in Figure 2.

6.       Other characterizations like XPS, specific surface area are required.

7.       Add a complete border to Figure 3 and 4. Unified style for all diagrams.

8.       In “3. Results and Discussion”, please write and discuss different test results in sections.

9.       The specific capacitance of FLG at 0.2A g-1 is calculated incorrectly using the equations (1).

10.   Please provide the calculation results of the internal resistance of the sample.

11.   Please pay attention to the writing of units. They should be written in the same style. “m2/g”, “mVxs-1” should be revised.

12.   The titles for Figure 3, 4, 10 are not right.

13.   1000 cycles are not enough to show the stability of carbonaceous materials. Please do more cycles.

 Moderate editing of English language is required.

Author Response

Reviewer 3

We are grateful to the reviewer for his attention to our publication. We have corrected all the comments and hope that our publication will be worthy of publication in “nanomaterials”. Corrections are marked in yellow in the text.

Below are the responses to the comments of the reviewer.

  1. The cited literature in the introduction section is mostly old, please add some latest literature to support it. Some typical references are suggested, e.g. Nanoscale 2022, 14, 8216; Materials & Design 2023, 229, 111904; Journal of Bioresources and Bioproducts 2022, 7 (4), 245-269.

Answer: We have expanded the introduction and used references recommended by the reviewer.

  1. Please introduce the sources of raw materials and reagents used in the experiment.

Answer: We have made the necessary changes to the text.

  1. It is recommended to add more SEM or TEM images to the article. The SEM image does not show a very intuitive few-layer structure.

Answer: We have added a TEM image, which clearly shows the few-layer structure of sample.

  1. Please unify the position for labeling in Figure 1 a and b.

Answer: We have redone the drawings in accordance with the comment of the reviewer.

  1. Label the vertical and horizontal coordinate scales in Figure 2.

Answer: We have redone the drawings in accordance with the comment of the reviewer.

  1. Other characterizations like XPS, specific surface area are required.

Answer: We have expanded the discussion of the results by discussing the specific surface area of a FLG.

  1. Add a complete border to Figure 3 and 4. Unified style for all diagrams.

Answer: We have redone the drawings in accordance with the comment of the reviewer.

  1. In “3. Results and Discussion”, please write and discuss different test results in sections.

Answer: We have expanded Results and Discussion section.

  1. The specific capacitance of FLG at 0.2A g-1is calculated incorrectly using the equations (1).

Answer:  We checked our calculations and found no error. Equation 1 was used for calculation capacitance values from CV. If we use equation 2 and the data from Figure 8, it would be (0.2 * 2348)/1.8=261 F g-1. The result in figure 8 and table 2 is slightly different because in the table and manuscript we used the average capacitance values for several parallel measurements.

  1. Please provide the calculation results of the internal resistance of the sample.

Answer: Since we performed the measurements in a three-electrode cell, the measurements of ESR are not quite correct, but in terms of ohmic resistance, from the EIS it was of the order of 100 Ohm. From the GCD measurements it was of the order of 60 Ohm. The rather high values are primarily related to the cell configuration and are primarily related to the electrolyte resistance (10.1002/aenm.201401401).

  1. Please pay attention to the writing of units. They should be written in the same style. “m2/g”, “mVxs-1” should be revised.

Answer: Thanks for your comment. We apologize for this oversight. This has been corrected in the new version of the manuscript.

  1. The titles for Figure 3, 4, 10 are not right.

Answer: We have corrected these typos.

  1. 1000 cycles are not enough to show the stability of carbonaceous materials. Please do more cycles.

Answer: For this type of materials we can expect a smooth linear decrease in capacitance, without noticeable changes. This is confirmed in many published papers for similar electrode materials (10.1016/j.apsusc.2022.153156 ; 10.1016/j.jelechem.2021.115626 ; 10.1002/admt.202101105 ; 10.1016/j.jallcom.2022.168568). We have made the necessary changes to the text.

Round 2

Reviewer 1 Report

The authors have carried a revision with an attempt to reply all questions from the reviewer. However, there are still some points which need clarification within the manuscript text prior to acceptance. 

  1. Page 14, Line 388-392: Authors have written: ‘As shown in [60; 61, 62, 63], the decrease in the capacity of supercapacitors based on carbon nanomaterials on the number of cycles has a linear dependence; therefore, at 5000 cycles, one can expect a decrease in the capacitance of a supercapacitor based on FLG to 80% of the initial value, which is a satisfactory value for applications in various electrical engineering.
  1. There are several assumptions regarding this explanation: Firstly, it is not clear relying on what the authors give that 80% capacitance loss of the initial value after only 5000 cycles could be a satisfactory value for various electrical engineering related applications. Secondly, this argument cannot be accepted if the performance of a supercapacitor containing the synthesized FLG electrode is not measured over 5000 cycles.  In this respect, the authors should revise by including a clearer explanation in  the manuscript.
  2. Authors response: ‘Since there are no pronounced peaks on CV, we used for calculations the values of currents on the plateau around 0.0 - 0.2 V, where the contribution of ohmic polarization does not have a significant effect on the CV shape’. It is recommended that the authors include this important observation in the revised manuscript (Page 10, Line 322-323) to make it clearer for readers.
  1. Page 11, Line 337-339: Authors have written: ‘All samples are characterized by deviations from the linearity of the charge-discharge curves, especially when discharging to low potentials, which is probably associated with side irreversible recovery processes.’ This statement is not clearly giving what is intended to mean: Is it meant by side irreversible recovery processes? It is required that the authors provide more clearly explained information including relevant references.

Authors response: ‘Most likely there is water splitting with the hydrogen evolution. Although the overpotential of hydrogen evolution at carbon electrodes reaches high values, but at potentials around -1.0 V vs Ag/AgCl and at low sweep rates/discharge currents the process can have an effect [10.1021/acsami.6b14732; 10.1021/acsami.7b06727]’.

Here too, it is advised that the the authors include these explanations or give references to support the explanations at the appropriate places in the revised manuscript for better understanding by the readers.

English language and style are fine, but minor spell check is required

Author Response

Please see in attached file.

Reviewer 2 Report

 Accept in present form.

Author Response

Thank you for your valuable comments earlier and for accepting the article now.

Reviewer 3 Report

The manuscript could be accepted now.

Author Response

(The authors gave the same response as above.)

Round 3

Reviewer 1 Report

Thank you for the appropriate revision.

Some expressions can be corrected by a native speaker-